# Study on the Application of Typhoon Experience Parameter Analysis in Taiwan's Offshore Wind Farms

Hui-Ming Fang [1], Hao-Teng Hsu [2] and Hsing-Yu Wang [3,*]

1. Bachelor Degree Program in Ocean Engineering and Technology, National Taiwan Ocean University, Keelung 20224, Taiwan; hmfang@mail.ntou.edu.tw
2. Ship and Ocean Industries R&D Center, Taipei 251401, Taiwan; cmhsu@soic.org.tw
3. Department of Shipping Technology, National Kaohsiung University of Science and Technology, Kaohsiung 805301, Taiwan
* Correspondence: hywang05@nkust.edu.tw; Tel.: +886-7-8100888 (ext. 25110)

**Abstract:** Due to the rapid development of computers, researchers have made efforts since the 1990s to develop typhoon forecasting models and stochastic typhoon simulation models to assess typhoon disasters and risks. Typhoon forecasting models are primarily used to predict and track the movement of typhoons and provide warning information to the general public before landfall. Stochastic typhoon simulation models can assess extreme wind speeds and compensate for the limitations of current observations and simulation data length. Taiwan experiences approximately three to four typhoons yearly, of varying intensities and paths. Whether the marine meteorological data includes events of strong typhoon centers passing through will affect the results of frequency analysis. The development of offshore wind power in Taiwan is closely related to the unique marine meteorological conditions throughout the lifecycle stages, including wind farm site selection, feasibility studies, planning and design, construction and installation, operation and maintenance, and decommissioning. This study references relevant research and analyzes sixty-three scenarios using nine types of maximum storm wind speed radii and seven Holland-*B* parameters. The data from Japan Meteorological Agency Best Track Data (JMA BTD) is utilized, explicitly selecting 20 typhoon events after 2000 for wind speed simulation using a typhoon wind speed model. After validating the typhoon wind speeds with observation data from the Central Weather Bureau (CWB) in Hsinchu and the Longdong buoy, the technique of Monte Carlo simulation is utilized to generate synthetic typhoons randomly. The average of the relative absolute errors for the simulated maximum wind speeds is calculated, and through comprehensive evaluation, optimal parameter combinations $(R_m, B)$ are obtained.

**Keywords:** typhoon forecast model; stochastic typhoon simulation model; Monte Carlo simulation; optimal parameter



## 1. Introduction

Taiwan is located along the path of typhoons in the northwest Pacific Ocean, and significant damage is caused by typhoons each year. Therefore, in the planning and design phase, the impact of typhoon wind speeds must be carefully assessed and analyzed. In general engineering design applications, assessing extreme wind speeds during typhoons is based on the observed wind speeds of typhoons and involves analyzing the regression of extreme values using probability density functions. However, most regions need long-term observation data for densely populated urban areas with long-term observation records. Furthermore, due to the expensive cost of measurements and the difficulty of maintaining instrument equipment on offshore structures, the measurement data is often too short of reliably describing the design wind speeds during typhoons. Therefore, establishing typhoon wind field models can address the abovementioned issues.

Numerical models that parameterize the wind speeds in the boundary layer of typhoons are currently widely utilized in research areas such as engineering design, numerical modeling, and risk assessment. These models play a crucial role in determining design standards for wind speeds, waves, sea currents, and storm surges in the design of offshore facilities. They are also applied in designing onshore infrastructure (including ports and building structures), disaster planning, personnel evacuation (considering wind and storm surge water levels), and risk assessment for insurance purposes. The first to utilize mathematical models to assess hurricane wind speeds along the Texas coast was [1]. Subsequent researchers expanded and improved upon the model [2–4], developing a model for typhoons' radial pressure and wind speed based on two empirical parameters: the maximum storm wind speed radius and the shape of the pressure field distribution. The model was validated by analyzing three Australian and nine Florida hurricanes. Reference [5] proposed a one-dimensional axisymmetric radial wind field model based on Monte Carlo simulation, verifying its accuracy using historical hurricane data. Ref. [6] employed hurricane track-tracking techniques to simulate the passage of each hurricane over the ocean and land. They derived typhoons' maximum wind speed radius by simulating and comparing them with key hurricane statistics observed along the Atlantic coast, including central pressure, hurricane translation speed, direction, and proximity distance. Reference [7] analyzed 842 typhoons between 1951 and 1997 with central pressures below 980 hPa. They statistically determined the average and standard deviation of the maximum storm wind speed radius and validated the impact on storm surge deviation along the Japanese coast by comparing it with observed values. For example, in analyzing the maximum storm wind speed radius ($R_m$) and Holland-*B* parameters, [8] employed a typhoon model to examine the maximum wind speed radius and Holland-*B* parameters, validating the analysis with observational data to explore the optimal parameter values for Shenzhen City. Reference [9] focused on the Zhoushan Archipelago, compiling previous studies on the maximum storm wind speed radius and establishing a typhoon model. They then validated the model using Typhoon Ampil (No. 1810) and Rumbia (No. 1817) as examples, comparing the results with data from three observation stations. They conducted error analysis using metrics such as mean bias, root mean square error, correlation coefficient, and scatter index to propose the parameter values most suitable for the maximum storm wind speed radius in the Zhoushan Archipelago.

In engineering applications, the scarcity of observational data often leads to increased statistical uncertainty. Therefore, the development of integrated typhoon simulation methods combining typhoon wind field models, probabilistic distributions of key typhoon parameters, and Monte Carlo simulation techniques has been pursued. Currently, widely accepted offshore wind turbine design guidelines and regulations in international practice are mainly based on experience from the offshore wind fields in the North Sea region of Europe. Design is typically carried out according to standards such as IEC (International Electrotechnical Commission) and DNVGL (Det Norske Veritas Germanischer Lloyd). Regarding tropical cyclone considerations, a 50-year return period is commonly adopted as the design extreme condition [10,11]. However, the development of offshore wind power in Taiwan is closely related to the unique marine and meteorological conditions throughout the lifecycle stages, including site selection, feasibility studies, planning and design, construction and installation, operation and maintenance, and decommissioning. Marine and meteorological data play a crucial role in achieving optimized development solutions that ensure both safety and cost-effectiveness. Taiwan is located in the low-latitude region of the western North Pacific, where it is prone to frequent typhoon impacts during the summer and autumn seasons. These severe marines and meteorological challenges imposed by typhoons are expected to impact offshore wind fields in Taiwan significantly. The successful development and operation of offshore wind farms in Taiwan depend on achieving a balance between safety and cost-effectiveness. This study references the methodology of surface wind field analysis [12] and involves compiling research on the maximum storm wind speed radius and Holland-*B* parameters. Validation uses observational data from

the Central Weather Bureau (CWB) Hsinchu and Longdong buoys. Monte Carlo simulation techniques are employed to generate synthetic typhoons, allowing for calculating the relative mean absolute error of the simulated maximum wind speeds. A comprehensive evaluation is conducted to determine the optimal parameter combination for a typhoon model that can be effectively applied in Taiwan.

## 2. Methodology

For the gradient wind speed model caused by typhoons, you can refer to the axisymmetric radial wind field model used by [5]:

$$u_G\left(\vec{x},t\right) = \frac{1}{2}(c \cdot \sin\alpha - f) + \sqrt{\frac{1}{4}(c \cdot \sin\alpha - f)^2 + \frac{B\Delta p}{\rho}\left(\frac{R_m}{r}\right)^B e^{[-(\frac{R_m}{r})^B]}} \tag{1}$$

where $u_G\left(\vec{x},t\right)$ is the gradient wind speed, $\vec{x} = (r,\theta)$ is the center of the polar coordinates located at the center of the typhoon, $f$ is the Coriolis force parameter, $\rho$ is the air density, $\alpha$ is the angle between the typhoon movement angle and the point (clockwise is positive), indicating the distance from the center of the typhoon to the target point, $c$ is the typhoon speed (km/h), $R_m$ is the maximum wind speed radius (km), and $B$ is the pressure profile parameter (generally called the Holland-$B$ parameter).

In addition, according to the research of [4], the typhoon radial pressure wind field can be expressed as follows:

$$p(r) = p_0 + \Delta p_0 e^{[-(\frac{R_m}{r})^B]} \tag{2}$$

where $p(r)$ is the air pressure at a radius (km) from the center of the typhoon, $p_0$ is the atmospheric pressure at the center of the typhoon (hPa), and $\Delta p_0$ is the air pressure difference between the center and the ambient pressure when the typhoon landfall (hPa). After substituting Equation (2) into Equation (1), Equation (3) can be rewritten as follows:

$$u_G\left(\vec{x},t\right) = \frac{1}{2}(c \cdot \sin\alpha - f) + \sqrt{\frac{1}{4}(c \cdot \sin\alpha - f)^2 + \frac{r}{\rho}\frac{\partial p}{\partial r}}. \tag{3}$$

Then, referring to the research of [13], the surface wind speed ($u_F$) and direction ($\theta_F$) can be expressed as Equations (4) and (5):

$$u_F\left(\vec{x},z,t\right) = u_G\left(\vec{x},t\right)\left(\frac{z}{z_g}\right)^{a_u}, \tag{4}$$

$$\theta_F\left(\vec{x},z,t\right) = \theta_G\left(\vec{x},t\right) + \gamma_s\left(1.0 - 0.4\frac{z}{z_g}\right)^{1.1}. \tag{5}$$

In Equations (4) and (5), the parameters of the formula are as follows:

$$a_u = 0.27 + 0.09\log(z_0) + 0.018\log^2(z_0) + 0.0016\log^3(z_0), \tag{6}$$

$$z_g = 0.052\frac{u_G\left(\vec{x},t\right)}{f_\lambda}(\log R_{0\lambda})^{-1.45}, \tag{7}$$

$$\gamma_s = (69 + 100\xi)(\log R_{0\lambda})^{-1.13}, \tag{8}$$

$$f_\lambda = \left( \frac{\partial u_G\left(\vec{x},t\right)}{\partial r} + \frac{u_G\left(\vec{x},t\right)}{r} + f \right)^{\frac{1}{2}} \left( 2\frac{u_G\left(\vec{x},t\right)}{r} + f \right)^{\frac{1}{2}}, \tag{9}$$

$$\xi = \frac{\left( 2\frac{u_G\left(\vec{x},t\right)}{r} + f \right)^{\frac{1}{2}}}{\left( \frac{\partial u_G\left(\vec{x},t\right)}{\partial r} + \frac{u_G\left(\vec{x},t\right)}{r} + f \right)^{\frac{1}{2}}} \tag{10}$$

where $z_0$ is the surface roughness length and $R_{0\lambda}$ is the Rossby number, which is used to represent the dimensionless ratio of the inertial force of the rotating fluid to the Coriolis force, which can be expressed as Equation (11):

$$R_{0\lambda} = \frac{u_G\left(\vec{x},t\right)}{f_\lambda z_0}. \tag{11}$$

In addition, refer to the study results of [14] as follows:

$$\rho = \frac{p_0 + \frac{\Delta p_0}{3.7}}{R T_{vs}}, \tag{12}$$

$$T_{vs} = (T_s + 273.15)(1 + 0.81 q_m), \tag{13}$$

$$T_s = 28 - \frac{3(\phi - 10)}{20}, \tag{14}$$

$$q_m = 0.9 \frac{3.802}{p_0 + \frac{\Delta p_0}{3.7}} e^{\frac{17.67 T_s}{243.5 + T_s}} \tag{15}$$

where $R = 286.9\,\text{J (kg\,°K)}$ is the gas constant of dry air, $T_{vs}$ is the virtual surface temperature, $T_s$ is the surface air temperature, $q_m$ is the vapor pressure at 90% relative humidity, and $\phi$ is the absolute latitude value. This study refers to the research results of [8,9], which use 9 types of $R_m$ and 7 kinds of Holland-*B* parameters; a total of 63 different combinations were analyzed. These combinations are described below.

### 2.1. Maximum Wind Speed Radius ($R_m$)

1. Applying hurricane trajectory tracking techniques, each hurricane is simulated as it traverses over the ocean and land. The maximum wind speed radius of a typhoon can be derived by simulating and analyzing key hurricane statistics observed along the Atlantic coast, including central pressure, hurricane movement speed, speed, heading, and approach distance [6].

$$R_m = \exp\left( 2.636 - 5.086 \times 10^{-5} \times \Delta p_0^2 + 3.94899 \times 10^{-2}\phi \right) \tag{16}$$

2. Using data from hurricanes spanning from 1900 to 1983, ground wind analyses from NOAA aircraft observations between 1995 and 2002, "Extended Best Track" data maintained by the National Hurricane Center (now with NOAA NESDIS located at Colorado State University) for the years 1988 to 1999, and HRD aircraft observation files for the years 1984 to 1987, the maximum wind speed radius can be established [15].

$$R_m = \exp\left[2.0633 + 0.0182 \times \Delta p_0 - 1.9008 \times 10^{-4} \times \Delta p_0^2 + 7.336 \times 10^{-4}\phi^2\right] \quad (17)$$

3.  From 1951 to 1997, a total of 842 typhoons occurred with central pressures below 980 hPa. The average and standard deviation of the maximum wind speed radius were calculated for these typhoons. These values were then verified and evaluated against observed data to assess their impact on storm surge deviations along the Japanese coast. Regression analysis was performed between the maximum wind speed radius and central pressure, resulting in the following two equations [7].

$$R_m = 80 - 0.769(950 - p_0), \quad p_0 \leq 950 hPa \quad (18)$$

$$R_m = 1.633 p_0 - 1471.35, \quad p_0 > 950 hPa \quad (19)$$

4.  Based on typhoon observation data along the Chinese coastline, a statistical analysis revealed the following relationship between the maximum wind speed radius and the central pressure difference for China's southern and eastern coasts [16].

$$R_m = \exp\left[-0.163 \times \Delta p_0^{0.555}\right] + 5.212 \quad (20)$$

5.  Statistical analysis of the $R_m$ model was conducted using flight-level and H*Wind data. These models were compared with models developed using more conventional methods [17].

$$R_m = \exp\left[3.015 - 0.00006291 \times \Delta p_0^2 + 0.0337\phi\right] \quad (21)$$

6.  Based on the analysis of observed data from the China National Typhoon Yearbook from 1949 to 2002, a recommended maximum typhoon wind speed radius is suggested [18].
$$R_m = 1.119 \times 10^3 \times \Delta p_0^{-0.805} \quad (22)$$

7.  The WRF-ARW model was used to simulate Typhoon Saomai, and validation was conducted using observational data from weather stations in Zhejiang Province, China. Based on this analysis, a recommended maximum typhoon wind speed radius is suggested [19].
$$R_m = \exp\left[-38.36 \times \Delta p_0^{0.025} + 46.75\right] \quad (23)$$

8.  The Federal Emergency Management Agency (FEMA) has developed a new statistical model that establishes the relationship between the maximum hurricane wind speed radius, hurricane center pressure, and latitude. This model has been used to update the maximum wind speed radius data for hurricanes Mitch (1998), Brett (1999), Floyd (1999), and Gilbert (1988) [20].

$$R_m = \exp\left[2.556 - 5.0255 \times 10^{-5} \times \Delta p_0^2 + 0.042243032\phi\right] \quad (24)$$

9.  Analyzing the data from the past 40 years, precisely ten typhoons that have impacted Ningbo City in mainland China, a relationship between the typhoon center pressure and the distance from the observation station has been established. This analysis has allowed for estimating the maximum hurricane wind speed radius for Ningbo City [21].

$$R_m = 29.178 \times \exp[0.0158 \times (p_0 - 900)] \tag{25}$$

### 2.2. Holland-B Parameter

1. The maximum hurricane wind speed radius can be determined by analyzing the observational data of typhoons passing over Lake Okeechobee in Florida. This analysis provides valuable insights into the impact of typhoons on the region and helps in assessing the potential risks and planning necessary measures for the area surrounding Lake Okeechobee [22].

$$B = 1.0 \tag{26}$$

2. By utilizing a typhoon model, a study can be conducted to analyze Tropical Cyclone Winifred. The model can simulate various aspects of the cyclone, including surface pressure and wind speed. Observational data from Cowley Beach in Australia can be used to validate the model's results to compare the simulated surface pressure and wind speed with the actual measurements. This verification process helps assess the typhoon model's accuracy and reliability in representing Tropical Cyclone Winifred's behavior [23].

$$B = 1.5 + \frac{(980 - p_0)}{120} \tag{27}$$

3. Holland-$B$ parameterization, as proposed by [24], suggests representing it as follows:

$$B = 2.0 - \frac{(p_0 - 900)}{160}. \tag{28}$$

4. Using data from NOAA-HRD and the Hurricane Reconnaissance Aircraft database, 201 profiles were examined with maximum wind speeds exceeding 33 m/s at flight altitudes below the 700 hPa pressure level. The analysis was conducted within the latitude range of 15° N to 35° N and the longitude west of 60° W in the Atlantic basin, resulting in the following findings [15].

$$B = 1.881093 - 0.005567 \times R_m - 0.010917 \times \phi \tag{29}$$

5. Using pressure data collected during hurricane reconnaissance flights from 1977 to 2001, the analysis was focused on data with pressures above 700 hPa. Additionally, H*Wind data for the hurricane wind field were analyzed. By incorporating the effects of maximum wind speed radius ($R_m$), central pressure difference ($\Delta p$), hurricane center latitude ($\phi$), and sea surface temperature ($T_s$), a new expression for the Holland $B$ parameter was derived [17]:

$$B = 1.7642 - 1.2098\sqrt{A}, \quad A = \frac{R_m \times f}{\sqrt{2R(T_s - 273) \times \ln\left(1 + \frac{\Delta p_0}{p_c \times e}\right)}} \tag{30}$$

where $T_s - 273 = 27\,°C$ is the sea surface temperature and $e$ is the Euler's number with a value of about 2.71828.

6. "A Revised Hurricane Pressure-Wind Model" proposes a novel technique that establishes a connection between the central pressure of a tropical cyclone and its maximum wind speed. It derives a new method for determining the Holland $B$ parameter, which is associated with the variations in pressure drop at the cyclone's center, the center's latitude position, and the tropical cyclone's translation speed [14].

$$B = -4.4 \times 10^{-5} \times \Delta p_0^2 + 0.01 \times \Delta p_0 + 0.03 \frac{\partial p_c}{\partial t} - 0.014\phi + 0.15 \times (v_t)^m + 1.0, \tag{31}$$

$$m = 0.6 - \left(1 - \frac{\Delta p}{215}\right) \tag{32}$$

where $v_t$ is the typhoon moving speed (m/s) and $\frac{\partial p_c}{\partial t}$ is the time-varying rate of central pressure difference intensity (hPa/h).

7. The typhoon parameters can be mutually verified by employing Monte Carlo hurricane simulation techniques and validating them using observed data from Typhoon Hagupit impacting Shenzhen City [8].

$$B = 0.8 \tag{33}$$

*2.3. Error Appraisal*

The wind speed values obtained from the typhoon model must be validated to demonstrate the model's accuracy. Error assessment methods, including mean bias error (Bias), root mean square error (RMSE), and scatter index (SI), are used to evaluate the discrepancies between the typhoon model and the observed data [9]. The formulas for these error assessment metrics are as follows:

$$Bias = \frac{1}{n}\sum_{i=1}^{n}(y_i - x_i), \tag{34}$$

$$RMSE = \sqrt{\frac{1}{n}\sum_{i=1}^{n}(y_i - x_i)^2}, \tag{35}$$

$$SI = \frac{1}{x}\sqrt{\frac{1}{n}\sum_{i=1}^{n}[(y_i - \overline{y}) - (x_i - \overline{x})]^2} \tag{36}$$

where $x_i$ is the $i$-th observation value, $\overline{x}$ is the average of the observation value, $y_i$ is the $i$-th simulation value, $\overline{y}$ is the average of the simulation value, and $n$ is the number of data.

The value of SI is about one order smaller than Bias and RMSE; so, to ensure the same weight, the value of SI will be multiplied by the weight coefficient $x$ and then the comprehensive error will be calculated.

$$Error = \frac{1}{3}\left(\overline{Bias} + \overline{RMSE} + xSI\right) \tag{37}$$

In wind speed error evaluation, they adopted $x = 10$ for calculation [9]. The symbol $^-$ in the above equations represents its mean value. The Bias term may have positive or negative values; so, in composite error calculations, the absolute values of the biases are summed. Additionally, in the simulation of typhoon wind speeds, the accuracy of the maximum wind speed simulation is also crucial, especially when using Monte Carlo methods for estimating wind speed regression values. Therefore, this study also includes an analysis of the relative error in maximum wind speeds during typhoon events. The formula for relative error in maximum wind speed is expressed as follows:

$$REMWS = \frac{[Max(y_i) - Max(x_i)]}{Max(x_i)} \tag{38}$$

where $Max(y_i)$ is the maximum wind speed calculated using the typhoon model and $Max(x_i)$ is the maximum value of the observed wind speed. If the relative error in maximum wind speed is positive, it means that the value calculated by the model is higher than the observed value. If the relative error in maximum wind speed is negative, it means that the value calculated by the model is lower than the observed value.

*2.4. Input Data for Typhoon Model*

The input data for the typhoon wind speed model is evaluated and analyzed using the Best Track Data (BTD) from the Regional Specialized Meteorological Center (RSMC) of the Japan Meteorological Agency (JMA). The determination of the typhoon's landfall time is based on the criteria provided by the CWB. The BTD database is widely recognized as one of the most reliable typhoon databases in the Northwest Pacific region and can be downloaded directly from the website. It is important to note that the time of the BTD data is not the same as the data provided by JMA. The BTD from JMA includes combination parameters with 3 h and 6 h intervals. These combination parameters include the time (in UTC), tropical cyclone, storm, and typhoon classifications, latitude and longitude coordinates of the typhoon center, central pressure intensity (in hPa), maximum sustained wind speed (in knots), minimum and maximum radii of 30 knots (15.43 m/s) and 50 knots (25.72 m/s) wind speeds, and corresponding wind directions. In applying stochastic typhoon models, parameters such as central pressure difference, maximum wind speed radius, typhoon translation speed, typhoon translation angle, and minimum distance to the typhoon are included. In synthesizing typhoon simulations using random numbers, it is assumed that the typhoon moves linearly according to its translation speed and angle.

**3. Simulation Results and Analysis**

The typhoon model in this study was verified using the wind speed data of Hsinchu and Longdong buoy of CWB. The information of the Hsinchu and Longdong buoy is shown in Table 1, and the buoy's location is shown in Figure 1. Since the time used by JMT data is UTC and the time used by CWB is UTC+8, special attention must be paid when converting data. The anemometers used in the buoys of the CWB have two types: 2 m elevation and 3 m elevation. Among them, there are more data at a 2 m elevation, so this study uses the wind speed data at a 2 m elevation for analysis. Then, for the convenience of verification, this study uniformly converted the wind speed to the 10 m elevation. Referring to [11], the conversion formula is as follows:

$$v(z) = v(2)\left(\frac{z}{z_2}\right)^{\alpha} \tag{39}$$

where $z_2 = 2$ m, $v(2)$ is the wind speed at a 2 m elevation (m/s) and $\alpha$ is the wind shear exponent. In the extreme wind speed model, $\alpha = 0.11$ is suggested.

**Table 1.** The brief descriptions of the buoy site.

| Name of Buoy | Coordinate | Water Depth | Closest Distance to the Coast |
|---|---|---|---|
| Hsinchu | 120.8436°, 24.7610° | 24.5 m | 6.4 km |
| Longdong | 121.9253°, 24.8483° | 21.0 m | 1.0 km |

In this study, typhoon data from CWB were selected, and typhoons after AD 2000 were used to verify the observation data and model. For those typhoons that have a slight impact on Taiwan and lack observation data, after elimination, they can be sorted into the typhoons in Table 2 for analysis. In Table 2, JMA No. is the typhoon number according to JMA BTD. The first 2 numbers are the last 2 digits of the AD year, and the last 2 numbers are the generation number of the typhoon in this year. The intensity of the typhoon is classified according to the CWB standard, and the classification is based on the maximum average wind speed in Table 3. Part of the track map of the verified typhoon is shown in Figure 2, and the source of the picture is CWB. In Figure 2, the colors of typhoon tracks represent typhoons of different intensities: blue line for mild typhoon, green line for moderate typhoon, and red line for severe typhoon.

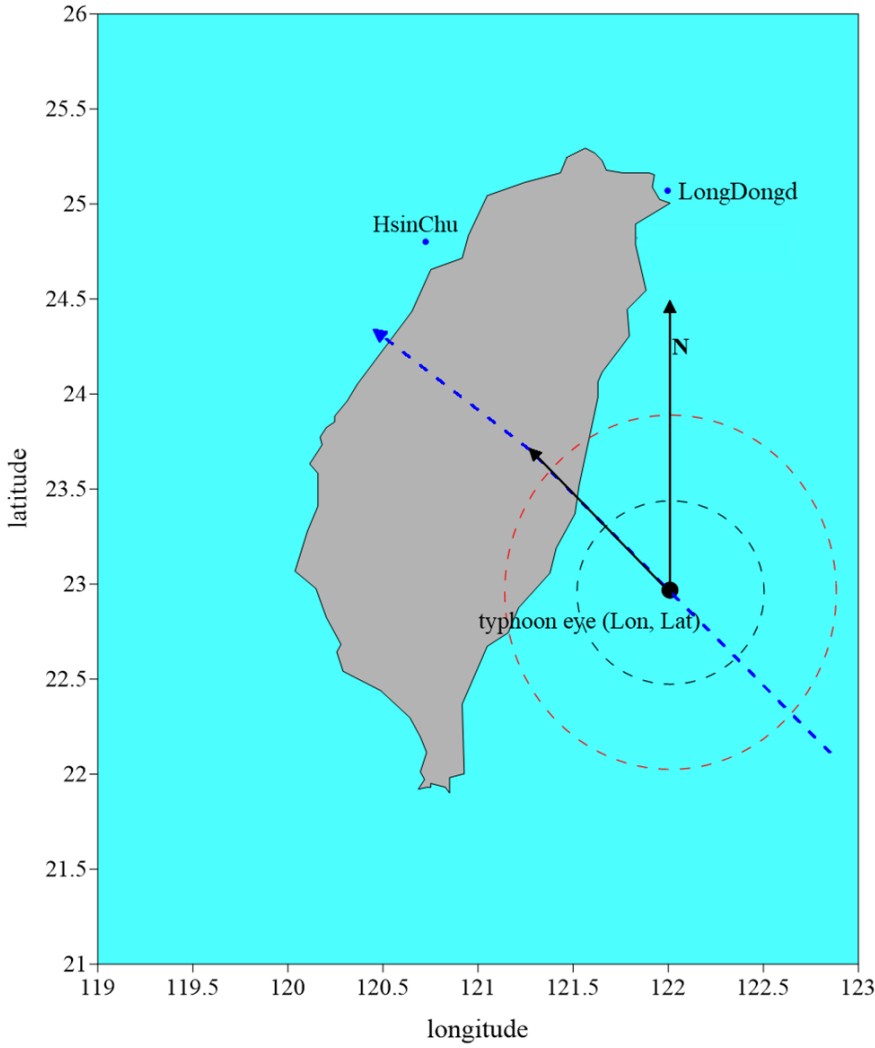

**Figure 1.** A schematic graph for the Hsinchu and Longdong buoy sites.

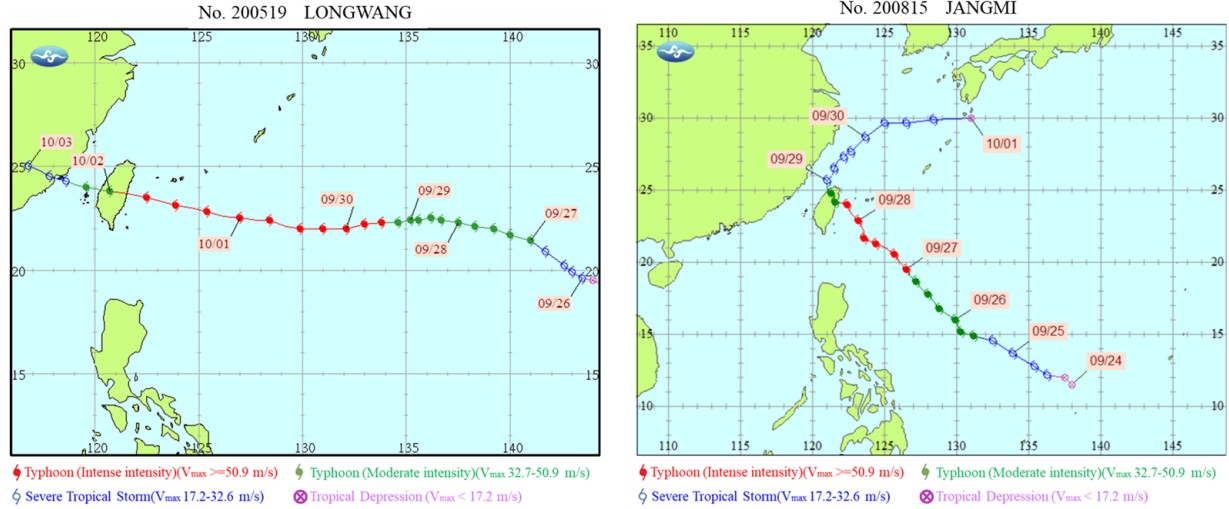

**Figure 2.** *Cont.*

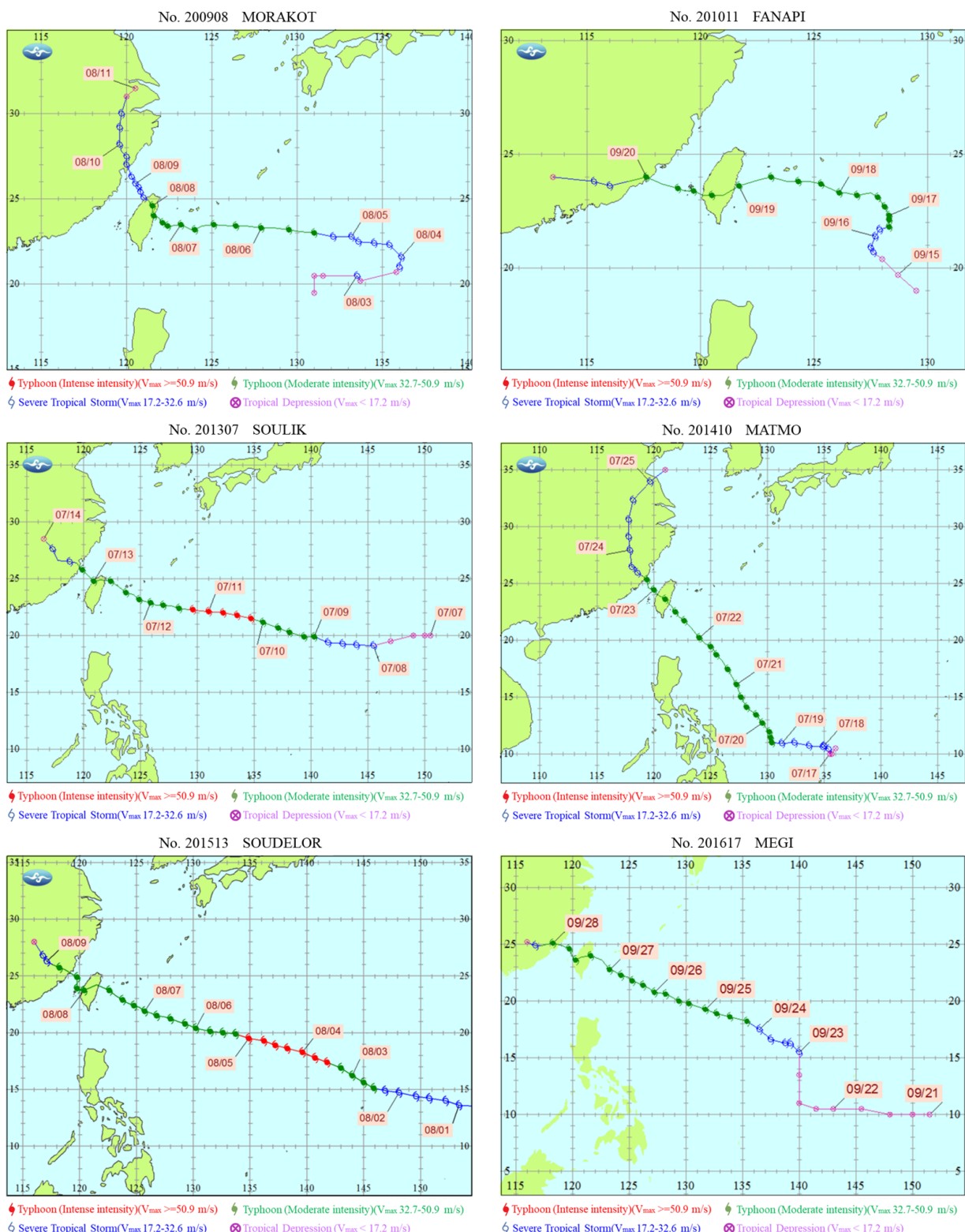

**Figure 2.** Part of the assessment typhoon track information and corresponding intensity.

**Table 2.** The typhoon information and corresponding intensity.

| JMA No. | Year | Typhoon | Category |
|---|---|---|---|
| 0208 | 2002 | NAKRI | Severe Tropical Storm |
| 0505 | 2005 | HAITANG | Typhoon (Intense Intensity) |
| 0513 | 2005 | TALIM | Typhoon (Intense Intensity) |
| 0519 | 2005 | LONGWANG | Typhoon (Intense Intensity) |
| 0604 | 2006 | BILIS | Severe Tropical Storm |
| 0708 | 2007 | SEPAT | Typhoon (Intense Intensity) |
| 0807 | 2008 | KALMAEGI | Typhoon (Moderate Intsnsity) |
| 0808 | 2008 | FUNG-WONG | Typhoon (Moderate Intsnsity) |
| 0813 | 2008 | SINLAKU | Typhoon (Intense Intensity) |
| 0815 | 2008 | JANGMI | Typhoon (Intense Intensity) |
| 0908 | 2009 | MORAKOT | Typhoon (Moderate Intsnsity) |
| 1011 | 2010 | FANAPI | Typhoon (Moderate Intsnsity) |
| 1111 | 2011 | NANMADOL | Typhoon (Intense Intensity) |
| 1209 | 2012 | SAOLA | Typhoon (Moderate Intsnsity) |
| 1307 | 2013 | SOULIK | Typhoon (Intense Intensity) |
| 1410 | 2014 | MATMO | Typhoon (Moderate Intsnsity) |
| 1513 | 2015 | SOUDELOR | Typhoon (Moderate Intsnsity) |
| 1617 | 2016 | MEGI | Typhoon (Moderate Intsnsity) |
| 1709 | 2017 | NESAT | Typhoon (Moderate Intsnsity) |
| 1710 | 2017 | HAITANG | Severe Tropical Storm |

**Table 3.** The wind speed of typhoon intensity near Taiwan defined by CWB.

| Intensity near Taiwan | Wind Speed near the Typhoon's Center |
|---|---|
| Severe Tropical Storm | 17.2~32.6 (m/s) |
| Typhoon (Moderate intensity) | 32.7~50.9 (m/s) |
| Typhoon (Intense intensity) | >51.0 (m/s) |

The typhoon model was verified using the observation data. Tables 4–8 are the analysis results with the Hsinchu buoy and Tables 9–13 are the results with the Longdong buoy. Because this study uses 20 typhoons for verification, for the convenience of reading and to evaluate the combined results of the maximum storm wind speed radius and Holland-*B*, the errors are averaged. In these tables, row 1 is the maximum storm wind speed radius used and column 1 contains the different Holland-*B* parameters.

**Table 4.** The Bias verified using CWB Hsinchu buoy data.

| $R_m$ \ $B$ | 1 | 2 | 3 | 4 | 5 | 6 | 7 |
|---|---|---|---|---|---|---|---|
| 1 | −3.26 | −5.61 | −5.49 | −4.89 | −5.51 | −3.47 | −2.74 |
| 2 | −5.62 | −8.26 | −8.16 | −7.88 | −8.36 | −4.80 | −4.71 |
| 3 | 1.29 | 0.79 | 0.93 | 0.73 | 1.19 | −1.18 | 0.72 |
| 4 | −8.87 | −10.97 | −10.92 | −10.92 | −11.16 | −7.10 | −7.73 |
| 5 | −2.36 | −4.48 | −4.36 | −3.65 | −4.29 | −2.99 | −2.03 |
| 6 | −0.93 | −2.45 | −2.30 | −1.75 | −2.11 | −2.47 | −0.96 |
| 7 | 1.02 | 0.48 | 0.64 | −0.71 | 0.80 | −1.46 | 0.49 |
| 8 | −3.31 | −5.67 | −5.55 | −4.96 | −5.57 | −3.50 | −2.79 |
| 9 | 0.44 | −0.47 | −0.33 | −0.05 | −0.07 | −1.68 | 0.08 |

**Table 5.** The RMSE verified using CWB Hsinchu buoy data.

| $R_m$ \ $B$ | 1 | 2 | 3 | 4 | 5 | 6 | 7 |
|---|---|---|---|---|---|---|---|
| 1 | 7.09 | 9.28 | 9.17 | 8.62 | 9.22 | 6.55 | 6.56 |
| 2 | 8.24 | 10.50 | 10.40 | 10.14 | 10.55 | 7.22 | 7.49 |
| 3 | 6.24 | 7.85 | 7.76 | 6.31 | 7.54 | 6.22 | 5.88 |
| 4 | 10.44 | 12.39 | 12.34 | 12.31 | 12.54 | 8.85 | 9.46 |
| 5 | 6.74 | 8.81 | 8.70 | 8.03 | 8.70 | 6.37 | 6.30 |
| 6 | 6.27 | 8.18 | 8.06 | 7.14 | 7.95 | 6.15 | 5.90 |
| 7 | 6.02 | 7.78 | 7.69 | 6.55 | 7.38 | 6.08 | 5.67 |
| 8 | 7.11 | 9.31 | 9.20 | 8.66 | 9.25 | 6.56 | 6.58 |
| 9 | 6.15 | 7.82 | 7.73 | 6.51 | 7.55 | 6.12 | 5.81 |

**Table 6.** The SI verified with CWB Hsinchu buoy data.

| $R_m$ \ $B$ | 1 | 2 | 3 | 4 | 5 | 6 | 7 |
|---|---|---|---|---|---|---|---|
| 1 | 0.48 | 0.56 | 0.56 | 0.54 | 0.56 | 0.43 | 0.45 |
| 2 | 0.47 | 0.50 | 0.50 | 0.50 | 0.50 | 0.42 | 0.45 |
| 3 | 0.45 | 0.59 | 0.59 | 0.47 | 0.56 | 0.46 | 0.43 |
| 4 | 0.44 | 0.46 | 0.46 | 0.45 | 0.46 | 0.42 | 0.43 |
| 5 | 0.48 | 0.57 | 0.57 | 0.54 | 0.57 | 0.43 | 0.45 |
| 6 | 0.46 | 0.58 | 0.58 | 0.52 | 0.57 | 0.43 | 0.43 |
| 7 | 0.45 | 0.59 | 0.59 | 0.49 | 0.56 | 0.45 | 0.42 |
| 8 | 0.48 | 0.56 | 0.56 | 0.54 | 0.56 | 0.43 | 0.45 |
| 9 | 0.46 | 0.59 | 0.59 | 0.49 | 0.57 | 0.45 | 0.43 |

**Table 7.** The comprehensive errors verified with CWB Hsinchu buoy data.

| $R_m$ \ $B$ | 1 | 2 | 3 | 4 | 5 | 6 | 7 |
|---|---|---|---|---|---|---|---|
| 1 | 5.15 | 6.88 | 6.78 | 6.36 | 6.81 | 4.79 | 4.74 |
| 2 | 6.17 | 7.93 | 7.86 | 7.66 | 7.97 | 5.41 | 5.57 |
| 3 | 4.49 | 5.54 | 5.50 | 4.49 | 5.36 | 4.42 | 4.22 |
| 4 | 7.89 | 9.32 | 9.27 | 9.26 | 9.41 | 6.71 | 7.17 |
| 5 | 4.83 | 6.47 | 6.38 | 5.85 | 6.37 | 4.62 | 4.48 |
| 6 | 4.41 | 5.84 | 5.75 | 5.04 | 5.64 | 4.45 | 4.17 |
| 7 | 4.33 | 5.46 | 5.41 | 4.58 | 5.22 | 4.35 | 4.07 |
| 8 | 5.17 | 6.90 | 6.81 | 6.39 | 6.84 | 4.80 | 4.75 |
| 9 | 4.40 | 5.44 | 5.37 | 4.60 | 5.28 | 4.39 | 4.17 |

**Table 8.** The relative error in maximum wind speed verified with CWB Hsinchu buoy data.

| $R_m$ \ $B$ | 1 | 2 | 3 | 4 | 5 | 6 | 7 |
|---|---|---|---|---|---|---|---|
| 1 | 0.28 | 0.42 | 0.41 | 0.38 | 0.43 | 0.28 | 0.25 |
| 2 | 0.35 | 0.42 | 0.45 | 0.44 | 0.46 | 0.33 | 0.33 |
| 3 | 0.20 | 0.40 | 0.39 | 0.21 | 0.37 | 0.19 | 0.17 |
| 4 | 0.55 | 0.69 | 0.69 | 0.69 | 0.71 | 0.48 | 0.50 |
| 5 | 0.26 | 0.39 | 0.38 | 0.34 | 0.39 | 0.26 | 0.23 |
| 6 | 0.21 | 0.31 | 0.31 | 0.25 | 0.31 | 0.23 | 0.18 |
| 7 | 0.19 | 0.36 | 0.36 | 0.20 | 0.34 | 0.19 | 0.17 |
| 8 | 0.28 | 0.42 | 0.41 | 0.38 | 0.43 | 0.28 | 0.25 |
| 9 | 0.19 | 0.35 | 0.34 | 0.22 | 0.33 | 0.20 | 0.17 |

**Table 9.** The Bias verified using CWB Longdong buoy data.

| $R_m$ \ $B$ | 1 | 2 | 3 | 4 | 5 | 6 | 7 |
|---|---|---|---|---|---|---|---|
| 1 | −3.19 | −5.73 | −5.58 | −4.92 | −5.58 | −3.26 | −2.66 |
| 2 | −5.56 | −8.34 | −8.22 | −7.89 | −8.38 | −4.62 | −4.64 |
| 3 | 1.65 | 1.35 | 1.47 | 1.27 | 1.71 | −0.79 | 1.00 |
| 4 | −8.79 | −10.99 | −10.93 | −10.88 | −11.11 | −6.93 | −7.64 |
| 5 | −2.27 | −4.60 | −4.44 | −3.65 | −4.36 | −2.76 | −1.92 |
| 6 | −0.84 | −2.64 | −2.48 | −1.70 | −2.27 | −2.19 | −0.84 |
| 7 | 1.38 | 0.92 | 1.06 | 0.10 | 1.26 | −1.05 | 0.79 |
| 8 | −3.25 | −5.79 | −5.65 | −4.99 | −5.65 | −3.29 | −2.70 |
| 9 | 0.69 | −0.28 | −0.14 | 0.30 | 0.12 | −1.34 | 0.30 |

**Table 10.** The RMSE verified using CWB Longdong buoy data.

| $R_m$ \ $B$ | 1 | 2 | 3 | 4 | 5 | 6 | 7 |
|---|---|---|---|---|---|---|---|
| 1 | 8.04 | 9.76 | 9.68 | 9.25 | 9.77 | 7.61 | 7.62 |
| 2 | 8.90 | 10.87 | 10.78 | 10.54 | 10.91 | 7.99 | 8.30 |
| 3 | 7.65 | 8.53 | 8.50 | 7.76 | 8.41 | 7.74 | 7.35 |
| 4 | 10.65 | 12.41 | 12.35 | 12.30 | 12.50 | 9.14 | 9.82 |
| 5 | 7.81 | 9.34 | 9.26 | 8.76 | 9.33 | 7.54 | 7.46 |
| 6 | 7.20 | 8.37 | 8.30 | 7.62 | 8.25 | 7.38 | 7.02 |
| 7 | 7.09 | 7.82 | 7.79 | 7.76 | 7.69 | 7.51 | 6.93 |
| 8 | 8.06 | 9.79 | 9.71 | 9.28 | 9.80 | 7.61 | 7.63 |
| 9 | 7.36 | 8.14 | 8.11 | 7.50 | 8.04 | 7.53 | 7.14 |

**Table 11.** The SI verified using CWB Longdong buoy data.

| $R_m$ \ $B$ | 1 | 2 | 3 | 4 | 5 | 6 | 7 |
|---|---|---|---|---|---|---|---|
| 1 | 0.57 | 0.58 | 0.58 | 0.59 | 0.60 | 0.56 | 0.56 |
| 2 | 0.55 | 0.53 | 0.53 | 0.54 | 0.54 | 0.53 | 0.54 |
| 3 | 0.58 | 0.66 | 0.65 | 0.60 | 0.64 | 0.63 | 0.56 |
| 4 | 0.48 | 0.47 | 0.47 | 0.47 | 0.47 | 0.48 | 0.50 |
| 5 | 0.58 | 0.59 | 0.60 | 0.60 | 0.62 | 0.57 | 0.56 |
| 6 | 0.54 | 0.56 | 0.56 | 0.55 | 0.57 | 0.57 | 0.53 |
| 7 | 0.54 | 0.62 | 0.62 | 0.62 | 0.61 | 0.61 | 0.53 |
| 8 | 0.57 | 0.58 | 0.58 | 0.59 | 0.60 | 0.56 | 0.56 |
| 9 | 0.56 | 0.61 | 0.61 | 0.57 | 0.61 | 0.60 | 0.55 |

**Table 12.** The comprehensive errors verified using CWB Longdong buoy data.

| $R_m$ \ $B$ | 1 | 2 | 3 | 4 | 5 | 6 | 7 |
|---|---|---|---|---|---|---|---|
| 1 | 6.39 | 7.50 | 7.48 | 7.27 | 7.63 | 5.61 | 6.04 |
| 2 | 6.71 | 7.80 | 7.77 | 7.69 | 7.90 | 5.88 | 6.33 |
| 3 | 5.92 | 6.25 | 6.29 | 5.83 | 6.33 | 5.85 | 5.74 |
| 4 | 7.73 | 8.81 | 8.78 | 8.79 | 8.92 | 6.63 | 7.19 |
| 5 | 6.27 | 7.36 | 7.34 | 7.03 | 7.49 | 5.61 | 5.99 |
| 6 | 5.70 | 6.37 | 6.34 | 5.85 | 6.34 | 5.53 | 5.60 |
| 7 | 5.40 | 5.68 | 5.71 | 5.72 | 5.71 | 5.62 | 5.33 |
| 8 | 6.39 | 7.51 | 7.49 | 7.27 | 7.65 | 5.61 | 6.04 |
| 9 | 5.74 | 6.03 | 6.04 | 5.66 | 6.04 | 5.64 | 5.65 |

**Table 13.** The relative error in maximum wind speed verified using CWB Longdong data.

| $R_m$ \ $B$ | 1 | 2 | 3 | 4 | 5 | 6 | 7 |
|---|---|---|---|---|---|---|---|
| 1 | 0.27 | 0.44 | 0.43 | 0.38 | 0.44 | 0.23 | 0.24 |
| 2 | 0.38 | 0.55 | 0.54 | 0.52 | 0.55 | 0.31 | 0.33 |
| 3 | 0.19 | 0.38 | 0.37 | 0.21 | 0.34 | 0.17 | 0.15 |
| 4 | 0.56 | 0.73 | 0.72 | 0.72 | 0.75 | 0.48 | 0.50 |
| 5 | 0.24 | 0.38 | 0.37 | 0.32 | 0.38 | 0.20 | 0.22 |
| 6 | 0.19 | 0.28 | 0.28 | 0.23 | 0.28 | 0.18 | 0.18 |
| 7 | 0.18 | 0.33 | 0.33 | 0.18 | 0.31 | 0.16 | 0.14 |
| 8 | 0.27 | 0.44 | 0.43 | 0.38 | 0.44 | 0.23 | 0.24 |
| 9 | 0.17 | 0.28 | 0.28 | 0.20 | 0.27 | 0.16 | 0.14 |

Tables 4–8 are the typhoon model's verification results and the Hsinchu buoy's observation data. Table 4 shows the verification results of Bias. The results show that, in different combinations of Holland-*B* parameters, [7,19,21] have more minor standard deviations in the maximum wind speed radius. The best combination is $R_m$ by [21] and *B* by [17], with an average standard deviation of −0.07. Table 5 shows the verification results of RMSE. The results show that the maximum wind speed radius of [7,19,21] in different Holland-*B* parameter combinations has a smaller RMSE. The best combination is $R_m$ of [19] and *B* of [8], with an average standard deviation of 5.67 (m/s). Table 6 shows the verification results of SI, and the overall results show little difference. The results show that [14,21,22] have better SI error in the parameter setting. Table 7 is the verification result of a composite error. The results show that, in different combinations of Holland-*B* parameters, [7,19,21] have a minor composite error in the maximum wind speed radius of the error. The best combination is $R_m$ of [19] and *B* of [8], and the composite error is 4.07. Table 8 is the verification result of the relative error in maximum wind speed, which calculates the relative error between the maximum wind speed of each typhoon simulation result and the observation data. The results show that a better result can be obtained in different combinations of Holland-*B* parameters, the absolute value of the maximum value of the simulated wind speed, after summing up the results and calculating the average. The results in Tables 4–8 show that the simulation results show that [7,19,21] have more minor errors. The $R_m$ of [8] has the most negligible error performance.

Tables 9–13 are the typhoon model's verification results and the Longdong buoy's observation data. Table 9 shows the verification results of Bias. The results show that, in different combinations of Holland-*B* parameters, [7,19,21] have more minor standard deviations in the maximum wind speed radius. The best combination is $R_m$ from [21] and *B* from [15], with an average standard deviation of 0.10. Table 10 shows the verification results of RMSE. The results show that [18,19,21] have smaller maximum wind speed radii under different Holland-*B* parameter combinations RMSE. The best combination is $R_m$ of [19] and *B* of [8], with an average standard deviation of 6.93 (m/s). Table 11 is the verification result of SI. Using [16]'s $R_m$ and different Holland-*B* parameters for simulation combination, the error results are not greater than 0.5. Table 12 shows the verification results of comprehensive errors. The results show that, in different Holland-*B* parameter combinations, [7,18,19,21] have a smaller maximum wind speed radius. The best combination is $R_m$ of [19] and *B* of [8], and the comprehensive error is 5.33. Table 13 shows the verification results of relative error in maximum wind speed. The results show that, in different combinations of Holland-*B* parameters, [7,19,21] have a smaller absolute value for the maximum wind speed radius of the relative error. The best combination is the $R_m$ of [19,21] with the *B* of [8], and the average error is 0.14. Using $R_m$ of [7] and *B* of [8], the average error is 0.15.

Then, use the $R_m$ of [19] with different Holland-*B* parameters, taking typhoon JANGMI as an example. The observation data and simulation results of the Hsinchu buoys are shown in Figure 3. The X-axis represents the time (UTC) and the Y-axis represents the wind speed

at an altitude of 10 m. The black dots in the figure are observation data, the black line is the simulation result using [22], the blue line is [23], the red line is [24], the black dashed line is [15], the blue dashed line is [17], the red dashed line is [14], and the black triangle is [8]. According to the typhoon warning issued by CWB, the warning time of typhoon JANGMI is 2008.09.26 15:00~2008.09.29 15:00 (UTC). As shown in Figure 3, the simulation results before 06:00 on the 28th are all smaller than the observation data, and only the simulation results of [8,18] are relatively close. In addition, the maximum wind speed simulation results show that the results are more accurate. The results of the other five Holland-*B* parameters are overestimated, and the worst occurs when using [23]. The observation data and simulation results of the Longdong buoy are shown in Figure 4. As shown in Figure 4, the simulation results before 00:00 on the 28th are smaller than the observation data, but the simulation results of [8,14] are relatively similar, followed by the results of [22]. In addition, the simulation results of the maximum wind speed show that the simulation results of [22] overlap with the observation data, the results of [8,14] are slightly lower, and the simulation results of the other five types are higher. In addition, as shown in Figures 3 and 4, the simulation results using [23,24] are almost overlapped. Using [17], it is slightly larger than the simulation results of [23,24] before the wind speed reaches the maximum value. However, the simulation results almost overlap until the wind speed gets smaller. Overall, using the $R_m$ of [19], the variation trends of the seven Holland-*B* parameters are similar. The simulation results of 20 typhoons were used in this study to verify the Hsinchu and Longdong buoys. The absolute values of the relative errors are 0.17 and 0.14, respectively, showing that the results are roughly consistent.

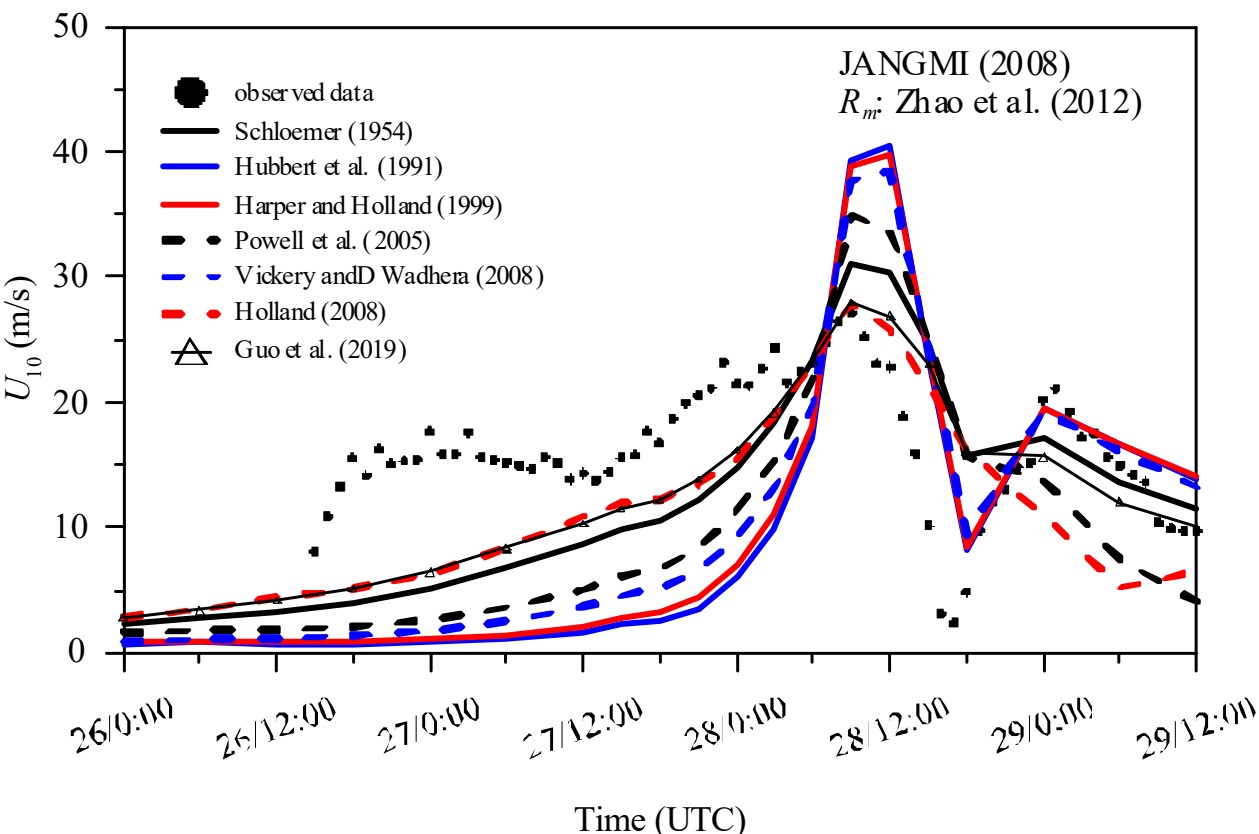

**Figure 3.** Time series of wind speed comparison between Hsinchu buoy data and computational results based on 7 Holland-*B* for JANGMI typhoon event [8,14,15,17,19,22–24].

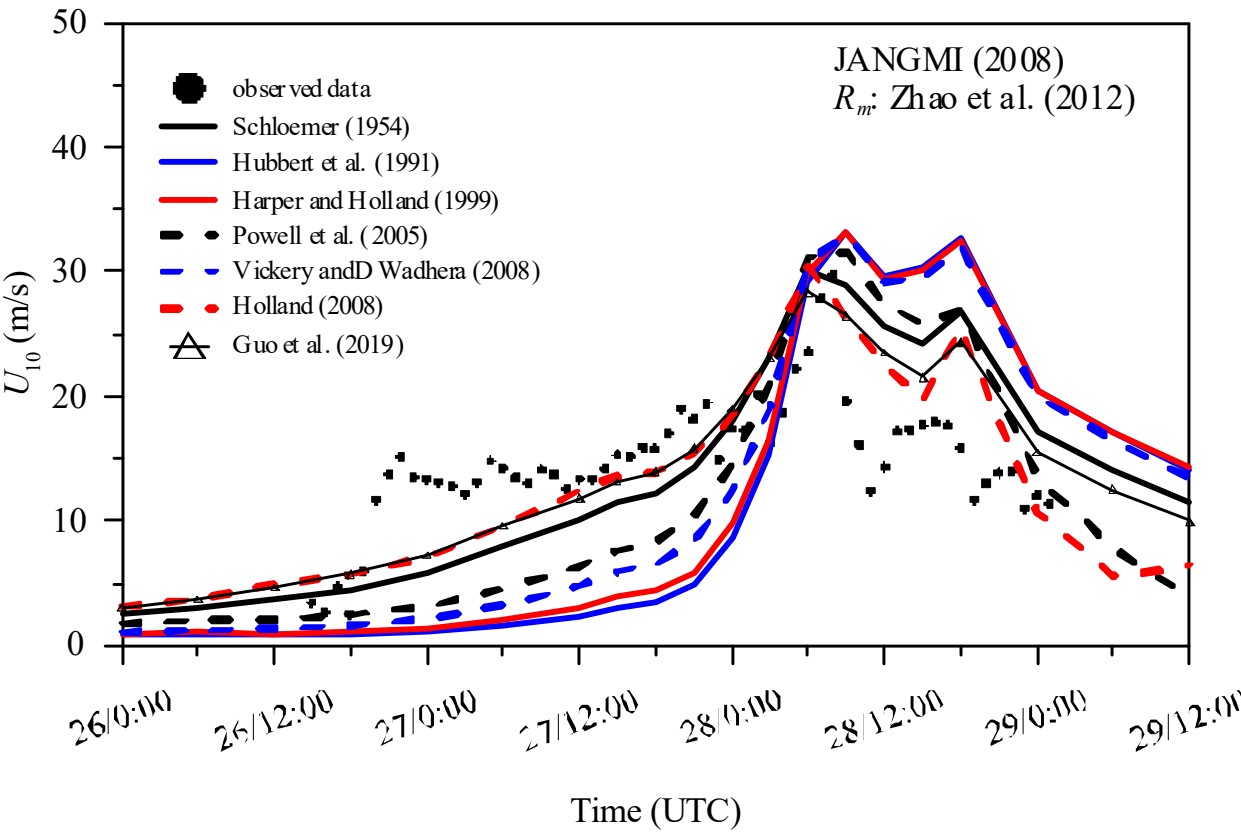

**Figure 4.** Time series of wind speed comparison between Longdong buoy data and computational results based on 7 Holland-*B* for JANGMI typhoon event [8,14,15,17,19,22–24].

## 4. Conclusions

In Taiwan, approximately three to four typhoons impact the region each year, with varying intensities and paths. Whether the marine meteorological data includes intense typhoon center passage events will affect the frequency analysis results. Taiwan is developing offshore wind power, and its various stages, including site selection, feasibility studies, planning and design, construction and installation, operation and maintenance, and decommissioning, are closely related to Taiwan's unique marine meteorological conditions. For the calculation of extreme wind speeds, it is recommended to use a stochastic typhoon model to simulate synthesized typhoons. This approach uses probability density functions to model the joint solution of five parameters, including central pressure difference, maximum wind speed radius, typhoon translation speed, typhoon translation angle, and minimum distance to the typhoon. This study aims to determine the optimal combination of parameterized typhoon models for simulating the Taiwan region. A total of sixty-three combinations were considered, comprising nine types of maximum storm wind speed radii and seven types of Holland-*B* parameter sets. The analysis in this study was conducted using the Japan Meteorological Agency (JMA) Best Track Data (BTD) for 20 typhoons that affected Taiwan from the year 2000 onwards. The verification was performed using meteorological observations from Hsinchu and Longdong buoy stations. Error analysis was performed using metrics such as mean bias, root mean square error, scatter index, and composite error.

Additionally, considering the application of Monte Carlo simulation for generating synthetic typhoons, particular attention was given to the relative absolute error in simulating maximum wind speeds. By comprehensively evaluating the results, an optimal combination of parameters was identified. Based on the verification analysis using the Hsinchu buoy data, it was observed that the suggested formulas by [7,19,21] for $R_m$ (maximum wind speed radius) provided better simulation results than the other six proposed

formulas. In the results of the Holland-*B* parameter, it can be observed that the results from [8,14,22] seem to have better error performance; based on the verification results from the Longdong buoy, it can be observed that in terms of the RMSE the recommended formula for $R_m$ performs best according to [19], followed by [21], which is slightly different from the results obtained from the Hsinchu buoy. The formula proposed by [18] ranks third in terms of performance.

Regarding the composite error performance, [19] exhibits the best performance, followed by the recommended formula [18,21]. Regarding the average absolute relative error in maximum wind speed, the $R_m$ formula by [21] shows the best performance, followed by the $R_m$ formula by [19]. The $R_m$ formula by [18] performs slightly worse than the previous two. Based on the comprehensive analysis, it can be concluded that using the $R_m$ formula by [19] or [21] in combination with B = 0.8 from [8] yields the lowest error. Taiwan has abundant wind energy resources, making offshore wind farm development highly advantageous. To calculate extreme wind speeds, it is recommended to use a stochastic typhoon model for synthesizing typhoons. In the future, when using Monte Carlo simulations to estimate typhoon wind speeds, the results of this research can be combined to establish a typhoon model for evaluating 50-year return wind speeds. This approach is expected to provide more accurate simulations of typhoon wind speeds.

**Author Contributions:** Conceptualization, H.-Y.W. and H.-M.F.; methodology and supervision, H.-T.H.; software, H.-M.F.; data curation, H.-Y.W. and H.-T.H.; validation, H.-T.H. and H.-M.F.; Writing—original draft preparation, H.-Y.W. All authors have read and agreed to the published version of the manuscript.

**Funding:** This research received no external funding.

**Institutional Review Board Statement:** Not applicable.

**Informed Consent Statement:** Not applicable.

**Data Availability Statement:** Not applicable.

**Conflicts of Interest:** The authors declare no conflict of interest.

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
