# Peer review of "Study on the Application of Typhoon Experience Parameter Analysis in Taiwan’s Offshore Wind Farms"

_water, doi:10.3390/w15142575_

Round 1

Reviewer 1 Report

To estimate the extreme wind speeds related to typhoon near Taiwan, some parameters of the stochastic typhoon wind fields simulation models were chosen by compering their performances using the observations of two buoy stations. This paper given the best choice of the parameters to estimate typhoon wind speeds near Taiwan when using Monte Carlo simulations. The results were clearly presented.

Minor revision

1. The buoy stations in caption of Figure 1 are "Hsinchu and  Longdong buoy sites", while those in the figure are "HainChu and Turtle Island". Should "Turtle Island" be "Longdong" ?

Author Response

Thank you for your guidance throughout the review process, and we will carefully address each comment and make the necessary revisions to improve the clarity, accuracy, and overall quality of the paper.

Reviewer 2 Report

Summary:

This manuscript describes research evaluating 63 combinations of the maximum wind speed radius and the Holland-B parameter, two key parameters used in typhoon wind speed models. These 63 parameter sets test different combinations of maximum wind speed radius and Holland B parameter formulations - expressed as functions of values such as minimum central pressure and typhoon center latitude that can be obtained from the Japan Meteorological Agency Best Track Data) proposed by previous studies. Near-surface buoy wind data from the Taiwan coast is used to test the accuracy of these maximum wind speed radius and Holland B parameters, the latter two of which are fed into a typhoon gradient wind speed model. The two selected buoys intercepted 20 past typhoon events. Using various error metrics, several maximum wind speed radius and Holland B parameter formulations are identified as the most accurate. This work is highly relevant for the Taiwan offshore wind power industry and other engineering applications because structures need to be built according to specifications appropriate for typhoon wind speeds expected in future storm events.   

Recommendation: Accept after Minor Revision

General comment:

Did you find any significant variability among the 20 different typhoon cases in terms of which (R_m, B) combinations were most appropriate? I am concerned that finding the optimal (R_m, B) parameters is not a "one size fits all" type of problem because of possible significant case-to-case variability. For example, intense typhoons tend to have much sharper radial profiles of the gradient wind speed compared to less intense typhoons or tropical storms.

Specific comments:

(1) Line 108: When describing Eq. (1), shouldn't "x" be a location somewhere in the typhoon wind circulation, not at the storm center? One would expect the gradient wind to be near zero at the storm center.

(2) Lines 112-113: Can you provide a bit more description of the Holland-B parameter? This is the radial surface pressure profile, right?

(3) Eq. 4: Should u{subscript_G} go in front of the first parentheses reading from the left?

(4) Line 127: Looking at your set of Equations, it looks to me like T{subscript_s} is surface temperature (units Celsius) while T{subscript_vs} is virtual surface temperature (units Kelvins). Please check.

(5) Table 2: Please change "Sever" in the right-most column to "Severe"

(6) Line 419: Shouldn't "RM" be written as Italicized R{subscript_m}?

(7) Can you provide the number of observation/simulation data points used in generating the statistics shown in Tables 3-12?

(8) Would it be possible to include any statistical significance testing for the error statistics evaluated in Tables 3-12? Showing statistically significant error reduction for some (R_m, B) combinations compared to others would strengthen your conclusions.

Author Response

(The authors gave the same response as above.)
